# A Simple Resident Need-for-Physical-Assistance Scale in Eldercare: Validation Using 4716 Observation Sequences of Caring Activities

**DOI:** 10.3390/ijerph191710488

**Published:** 2022-08-23

**Authors:** Sandra Schade Jacobsen, Matthew Leigh Stevens, Kristina Karstad, Charlotte Diana Nørregaard Rasmussen, Alexander Bork Kühnel, Andreas Holtermann

**Affiliations:** The National Research Centre for the Working Environment (NRCWE), Lersø Parkallé 105, 2100 Copenhagen, Denmark

**Keywords:** musculoskeletal disorders, physical demands, workplace

## Abstract

Accurate and simple measures for classifying nursing home residents according to their care needs would be valuable for planning eldercare work. Our aim was to validate a developed classification scale of residents’ need for physical assistance. Eldercare workers and managers in 20 Danish nursing homes classified 1456 residents into four categories (from light to complete need for physical assistance). We validated the resident need-for-assistance scale against 4716 workplace observation sequences of caring activities performed by eldercare workers. We found a strong correlation between the resident need-for-assistance scale and observed number of resident handlings (*r* = 0.71) and a moderate correlation for observed duration of care (*r* = 0.57). The discriminative ability of the scale was good for both number of resident handlings (ROC-AUC = 0.81) and for duration of care (ROC-AUC = 0.76). Our findings indicate that this simple scale is valid and feasible for classifying residents according to their physical assistance needs.

## 1. Introduction

Workers in eldercare have a high prevalence of musculoskeletal pain and long-term sickness absence, as well as an increased risk of early withdrawal from the labor market [1,2,3]. We know that these adverse health-related outcomes are associated with the eldercare workers’ physical demands in caring for the residents [4,5]. The physical demands put on the eldercare workers are predominantly determined by the need for physical assistance of the residents they care for [6,7]. However, we know little about how to distribute the residents with diverse physical assistance needs between workers within teams in eldercare. Thus, some workers might be allocated more of the residents with greatest need for physical assistance, giving them a very high physical work demand, and consequently an increased risk for adverse health-related outcomes. Therefore, we need more information about the best way to allocate and distribute residents to the workers to make the best fit.

For this purpose, no simple, feasible and validated scale exists for determining the individual residents’ need for physical assistance. Existing systems, such as the Care Thermometer, RAI or MAPO Method, provides risk assessment in the manual handling of residents, primarily aiming to estimate the physical function of the resident or pinpoint which ergonomic equipment and/or preventive measures are needed in caring for the residents [8,9,10,11,12]. While this is also useful, these systems are quite extensive and burdensome for regular use. Moreover, they do not give the information needed for daily distribution of residents, with different physical assistance needs, between workers within teams in eldercare. Nonetheless, these existing systems have not been validated against observations of the residents’ physical assistance needs, but against self-registration or other assessment tools [10]. Thus, we need a simple, feasible and validated tool to assess the resident’s need for physical assistance in eldercare.

In Denmark, many eldercare facilities use a simple, single item, four- or five-point categorization scale for quantifying the individual resident’s need for physical assistance. Since many of these scales have been developed by the nursing homes themselves, they lack standardization and validation. To deal with this, we have developed a simple four-point scale to be used in 20 Danish nursing homes participating in the Danish Observational Study of Eldercare work and musculoskeletal disorderS (DOSES) [13]. Our aim of this study was to validate this simple four-point resident need-for-assistance scale against observed physical work demands of the workers caring for the residents.

## 2. Materials and Methods

This validation study was conducted using baseline data from the DOSES cohort [13]. DOSES is a prospective workplace observational study designed to examine longitudinal associations between physical and psychosocial working conditions and occurrence of musculoskeletal disorders and its consequences among eldercare workers in Danish nursing homes. DOSES received ethical approval from the Danish Data Protection Agency and the Ethics Committee for the regional capital of Denmark (H-4-2013-028). All parties involved in the study (i.e., workers and residents) gave informed consent for the collection of data.

### 2.1. Study Population

A total of 83 nursing homes located in the Copenhagen area were invited to participate in DOSES. The nursing homes were purposively selected with the aim to include various care models (e.g., dementia and somatic units) and sizes of nursing homes. In short, the nursing homes were invited to participate in the study by direct contact by email and follow-up phone calls to the management (for further recruitment details see Karstad et al., 2018 [13]). Twenty nursing homes agreed to participate and were subsequently included in DOSES. Two nursing homes were private and 18 were public nursing homes. Of the total 126 wards, 92 were primarily somatic units, 28 dementia units, 3 temporary/rehabilitation units and 3 psychiatric units for caretaking of elderly people with special needs. The 20 nursing homes had on average 6.3 wards (SD 3.1), 79 residents (SD 28.9) and 70 workers (SD 27.7) employed for more than 15 h per week on day and evening shifts. Participation in the direct observations of this study was voluntary for eldercare workers and residents. The data collection in the DOSES cohort was carried out from September 2013 to January 2016.

### 2.2. Study Population

#### 2.2.1. Observational Data

To collect data on the physical assistance required to care for each resident, we used direct observations to capture the caring activities performed by the eldercare worker when looking after each resident. These real-time workplace observations took place at each nursing home over one to two weeks using a specific protocol based on previous studies [14,15,16,17,18,19]. Observations were collected by trained personnel (18 students or graduate students from various health educational programs) using tablets containing the software “Noldus Observer XT pocket observer” (Noldus, Wageningen, The Netherlands). The inter-rater reliability of the observation instrument has been demonstrated to be good, and is described elsewhere [20]. The observations were collected for each resident during a single day shift and a single evening shift within the same week, and was mostly conducted by two different observers (one observing during the day shift and one during the evening shift). The observers performed observations of all work tasks related to care of the residents during day shifts (4-h observation period) and an evening shifts (4–5-h observation period). The observations were conducted with the aim of minimal interference with the eldercare work, and without knowledge of the resident need-for-assistance classification. The observers collected information on all main work tasks related to care of the residents. These included (1) type of activity or routine, (2) resident handling, (3) if and what type of ergonomic devices were used during handling activities, (4) help from colleague or others during resident handling activities, (5) residents’ self-reliance during resident handling activity, (6) barriers for carrying out the work task and (7) emotional demands from resident. This study uses the observational data on the total duration of care (the total time the eldercare worker spent in the resident’s room), the total number of handling activities, individual handling activities (lifting, repositioning and turning of the resident) and handling activities performed without support from the resident.

#### 2.2.2. Development of the Scale and Classification of Residents

The resident need-for-assistance scale was developed in collaboration with workers and managers from several nursing homes. During the early stages of DOSES, we found that some nursing homes, but not all, used tools for categorizing the residents in their daily work. However, the tools were different and were used differently between the nursing homes. We therefore saw a potential to develop one simple and validated tool for categorizing residents, which could be used in planning and allocating residents between workers in a team.

In order to ensure content validity, we involved the practitioners in the development of the scale, so that the content of the scale was an adequate reflection of the construct (the resident’s need for assistance) to be measured. We made a first draft of the scale in April 2013 based on three examples collected from three different nursing homes located in the greater Copenhagen area. We aimed to keep the scale close to the existing scales used by the nursing homes to enhance the recognizability and facilitate its practical use. In August 2013, we further collected verbal feedback on the first version of the scale from four managers at different nursing homes, specifically on the usefulness and understandability. They were asked (1) whether the scale was easy to understand and (2) whether it was possible to categorize their residents using the scale. Based on their feedback, we made small adjustments to the scale, e.g., expanded the scale from three to four categories and made changes in the wordings.

The need-for-assistance scale classifies residents into four categories based on the resident’s need for physical assistance. The definition of each category is presented in Figure 1. The eldercare workers and/or managers conducted the classification of residents during a 3-week period around the same time as the observations. Based on their knowledge of the residents, the workers and/or managers categorized each resident based on the need-for-assistance scale and registered the classification in a table provide by the researchers.

### 2.3. Validation Process and Statistical Analysis

This validation was conducted in line with the recommendations of the COnsensus-based Standards for the selection of health Measurement INstruments (COSMIN) group [21]. To validate the resident need-for-assistance scale, we assessed its criterion validity and discriminative ability against the observed physical assistance needs. Thus, we compared the scale’s ratings (light physical assistance, moderate physical assistance, extensive physical assistance and complete physical assistance of the 1456 residents—based upon the appraisal of the eldercare worker conducting the assessment) to the 4716 observations of handling activities and duration of care. Our primary outcomes were the average observed number of handlings and average observed duration of care required over a whole day (day and evening shift combined). Secondary outcome measures were the observed individual handling activities (the number of lifting activities, turning activities and repositioning activities) required over a whole day, and the observed number of handling activities conducted without support from the resident over a whole day. As there are large differences in the assistance required between day and evening shifts, we conducted the analysis stratified by the type of shift (day or evening shift) as well.

Initially, we explored the observational data by conducting descriptive analyses and visual comparisons of the resident need-for-assistance scale and observed need for physical assistance using boxplots. To assess criterion validity, we used the Spearman’s rho test analyzing the correlation between the resident need-for-assistance scale and the observed need for assistance. In accordance with Schober et al. 2018, we interpreted the correlation coefficients as: ≥0.90 being very strong, 0.70–0.89 as strong, 0.40–0.69 as moderate, 0.10–0.39 as weak, and <0.10 as negligible [22]. Confidence intervals for Spearman’s rho were obtained through bootstrapping using 1000 replicates.

To assess the discriminative ability (the scale’s ability to differentiate with respect to the outcome) of the resident need-for-assistance scale, we conducted a receiver operating characteristics (ROC) analysis for all outcomes. The area under the ROC curve (ROC-AUC) provides a measure of classification accuracy based on the residents’ observed need for physical assistance. The predefined interpretation of the ROC-AUC values were as follows: values of 0.97–1.00 are identified as excellent, 0.93–0.96 as very good, 0.75–0.92 as good, <0.75 has obvious deficiencies, and <0.50 indicates that the test has no discriminative ability [23,24].

All statistical analyses were conducted using R v4.0.2 (R Foundation for Statistical Computing, Vienna, Austria) [25]. The R-packages that were used for analysis were ‘haven’ [26], ‘dplyr’ [27], ‘pROC’ [28] and ‘RVAideMemoire’ [29] packages.

## 3. Results

A total of 1456 residents were observed (88% and 79% of total number of residents during day and evening shifts, respectively). Across the 20 included nursing homes, we recorded a total of 4716 (2673 day and 2043 evening) observation sequences of direct caring activities (an interaction between an eldercare worker and a resident). Possible reasons for not observing some sequences of direct caring were if the resident was not present at the nursing home (e.g., due to visiting the family) or if the resident or the eldercare worker did not give permission for us to perform the observation. Of the 126 participating wards, a total of 120 wards classified their residents by filling in and returning the scheme (95.2%). The 120 wards had 1355 residents, of which 1317 residents were classified (97.2%) according to the resident need-for-assistance scale.

The descriptive results are presented in Table 1, showing that the residents had a similar body weight, but varied greatly in their observed need for assistance. The number of handlings (Figure 2A) for residents categorized as in need of light and moderate physical assistance were nearly zero, whilst residents categorized as in need of complete assistance had an average of eight handlings per day. The latter consisting of approximately two lifting activities, two repositioning activities and three turning activities, without supporting the handling by themselves (six numbers of handlings without support from the resident). The duration of care (Figure 2B) ranged from less than 30 min of care per day (for residents categorized as in need of light physical assistance) to more than one hour (for residents categorized as in need of complete physical assistance).

Descriptive results for day and evening shifts are presented in the Appendix A (Table A1). The majority of care activities took place during the day, ranging from nearly zero to five resident handlings, and 17 to 48 min of care. During evening shifts, resident handlings ranged from nearly zero to two, and 6 to 18 min of care.

### 3.1. Criterion Validity

We found a positive and strong correlation between the resident need-for-assistance scale and observed number of handlings with the Spearman’s rho correlation coefficient 0.71 (95% CI 0.67–0.74). The correlation between the resident need-for-assistance scale and observed duration of care was moderate, with a correlation coefficient of 0.57 (95% CI 0.52–0.61). The results were similar for three of the secondary outcomes, finding a moderate correlation between the scale and observed number of lifting activities (0.63 (95% CI 0.59–0.66)), repositioning activities (0.52 (95% CI 0.47–0.57)) and turning activities (0.61 (95% CI 0.57–0.65)). The correlation between the resident need-for-assistance scale and observed number of handlings without support from the resident was strong with a correlation coefficient of 0.71 (95% CI 0.67–0.73). Correlation coefficients between the resident need-for-assistance scale and the designated outcomes are listed in Table 2. No clear differences were found between day and evening shifts (Appendix A, Table A2).

### 3.2. Discriminative Ability

The overall ROC-AUC values between the resident need-for-assistance scale and the observed number of handlings and observed duration of care were 0.81 and 0.76, respectively, indicating that the resident need-for-assistance scale has a good discriminative ability. This was similar for the observed number of handlings without support from the resident, demonstrating a ROC-AUC value of 0.78. All individual handling activities demonstrated rather poor discriminative ability with values ranging from 0.71 to 0.74 for the number of lifting, turning and repositioning activities. The overall ROC-AUC values for all outcomes are listed in Table 2, while the individual ROC curves (presenting the ROC curves comparing each classification category to all the others) for the primary outcomes are shown in Figure 3.

The findings were similar when we included day shifts only, which showed that observed duration of care, number of handlings and handlings without support from the resident had a good discriminative ability, while individual handling activities had a poor discriminative ability. The discriminative ability for all outcomes during evening shifts demonstrated minor deficiencies (see Appendix A, Table A2).

## 4. Discussion

Our study aimed to validate a simple classification scale of residents’ need for physical assistance in nursing homes. Overall, we found that the resident need-for-assistance scale was valid in discriminating between the observed need for physical assistance of the residents. To the best of our knowledge, this is the first study to validate a simple and feasible resident need-for-physical-assistance scale against workplace observations of daily caring activities in nursing homes.

Our first main finding was the positive correlation between the resident need-for-assistance scale and the observed number of handlings (Table 2). This means that the number of handlings required to care for the resident increases with the need-for-assistance classification. This is in line with previous findings demonstrating that a high assistance need among residents is associated with the number of resident handlings performed in a shift [7]. We found a similar correlation between the resident need-for-assistance scale and the observed number of handlings performed without support from the resident, demonstrating an increasing need for physical support with an increasing need-for-assistance classification. The correlation was lower for the individual handling activities (lifting, repositioning and turning). This was expected, since the intention of the scale is not to specify which specific type of physical assistance the resident needs, but rather the overall need for physical assistance.

Our second main finding was the positive moderate correlation between the resident need-for-assistance scale and the observed duration of care (Table 2). This means that the time required caring for the residents increases with the need-for-assistance classification. Because the time spent caring for a resident not only depends on the physical assistance need of the resident, but also other needs (e.g., chatting and socializing with the resident), we expected the correlation to be somewhat lower than for the observed number of resident handlings. However, in a practical perspective, residents classified as in higher need for physical assistance seem to require more time to care for throughout the day.

Overall, we find these correlations to be quite strong considering that the evaluation was performed in a natural work setting across many nursing homes in Denmark. First, compared to other existing tools [10,11,12] the resident need-for-assistance scale is based on a simple item, which was feasible for the eldercare workers and/or manager to fill out during daily work (95% of the included wards filled in the classification schemes). Second, the classification of the resident was not conducted on the same day as the observations, but in a three week period of the day of observation. Thus, it seems that this simple scale can be used in daily practice during eldercare work, giving a good indication of the actual need for physical assistance of the residents.

We further evaluated the scale’s ability to distinguish between the different categories of the resident’s need for physical assistance. The ability was found to be good for both the observed number of handlings and observed duration of care (Table 2).

Furthermore, we compared each classification category against each other to evaluate their separability (Figure 3). According to our predefined interpretations, the ability to distinguish between the classification categories was overall good. Light compared to moderate need for physical assistance demonstrated the lowest separability in both the number of handlings and duration of care. This indicates that it might have been more difficult for the eldercare workers and managers to classify the residents in these two categories of the scale. It is likely that residents in these two lower categories (light and moderate need for assistance) exhibit a greater day-to-day variation in their physical abilities, making it more difficult to classify them. Due to the low discriminability, it may seem rational to merge the two lower categories into one, thus reducing the scale to three categories as the original draft of the scale. However, feedback from the nursing homes revealed that four categories was more feasible and made more sense for the eldercare workers and managers in their daily work. With the overall interpretation that the scale differentiates well between the need for physical assistance of the residents as well as being applicable to practice, we think that the scale should be kept at four categories.

### 4.1. Strength and Limitations

The main strength of this study is the large sampled data of systematic and reliable workplace observations from 20 nursing homes and 126 wards. The participating nursing homes represent a broad spectrum of nursing homes in Denmark including residents with a wide variety of conditions, which is a strength for the validation. The strength of our study further lies in its close relation to practice. The scale is simple and feasible and can therefore be used in the daily distribution of the residents with different needs for physical assistance within teams of eldercare workers. Furthermore, eldercare workers are often transient with significant turnover [30,31]. Simple and feasible tools, such as ours, may help to improve patient care by providing clear direction to staff who may be unfamiliar with resident needs.

The study also had some limitations. First, the classification and observations were not performed at the same day. The results can be expected to be even more in favor of being valid if the classification and observation was performed at the same day. Moreover, by not assessing confounding factors such as eldercare characteristics (e.g., seniority) it is possible that bias may have been introduced. Another limitation is that we did not adjust for residents’ characteristics nor do we have the data to analyze the variations in needs for assistance within residents. Additionally, it is a limitation that our scale only takes the physical need for care into account. We acknowledge that care also involves psychosocial factors, which can be important for the social relation between individual residents and eldercare workers and the mental load on the eldercare worker caring for the resident. Future studies ought to investigate whether a similar scale for the resident psychosocial care needs is valid for quantifying the psychosocial burden and time use of care.

### 4.2. Practical Implications for Eldercare Work

Our finding of a rather high correlation between the resident need-for-assistance scale and the observed need for assistance, as well as separation between the scale’s classification categories, indicates that the scale is valid. Furthermore, eldercare workers and managers classified as much as 97% of their residents, indicating that the scale is feasible. Despite the scale being developed in 2013, there has been no major changes within the eldercare sector (e.g., resident population, organization of eldercare, education, quality and performance of care, or resource allocation) in Denmark since. We therefore have no reason to believe that the time passed since the evaluation of the scale will have any significant effect on its generalizability, applicability, usability, reliability and validity in eldercare in Denmark. Thus, this simple resident need-for-assistance scale can be used during normal eldercare work, but we recommend researchers and practitioners of other countries and contexts to evaluate its usability and validity in their specific context, since it has been developed to be used and fit in Danish eldercare. In addition, the scale can be useful in care planning of residents with different needs for physical assistance between eldercare workers within a team. In a practical everyday setting, this means that when the workers and/or managers plan the workday and distribute the residents among the workers in the team, the categorizations of the resident’s need for physical assistance will aid in avoiding one employee having a large proportion of physically demanding residents. If this is the case, they will thus modify the plan to more evenly distribute the residents with highest need for physical assistance among the workers. Thus, the scale might be valuable in preventing a high physical workload on some eldercare workers, and consequent musculoskeletal pain and sickness absence [32].

## 5. Conclusions

In conclusion, our findings of a high correlation between the resident need-for-assistance scale and the observed need for assistance, and separation between the scale’s categories, indicate that this simple resident need-for-assistance scale is valid and feasible for classifying residents according to their need for physical assistance. Because this validation was performed across 20 nursing homes, 1456 residents and their caregivers during normal daily work, we believe that this simple scale can be used in the daily practice of eldercare work, and potentially be valuable in the planning of care of residents with different needs for physical assistance between eldercare workers within a team.

## Figures and Tables

**Figure 1 ijerph-19-10488-f001:**
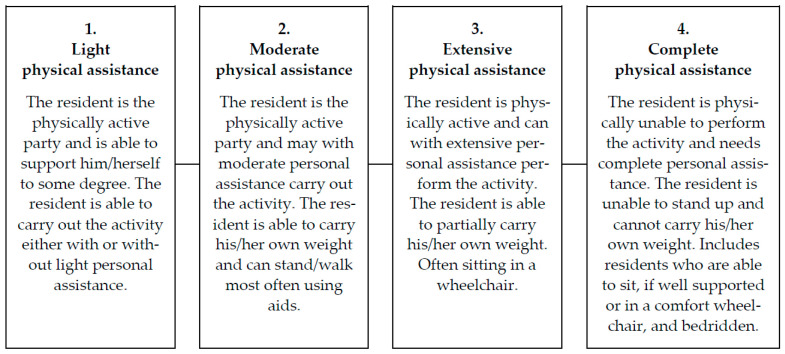
The resident need-for-physical-assistance scale.

**Figure 2 ijerph-19-10488-f002:**
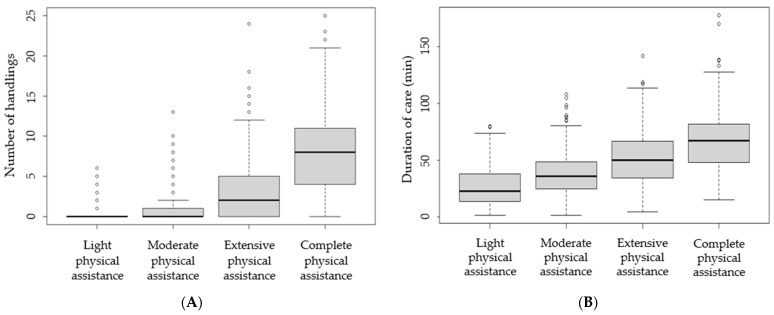
Boxplots for (**A**) total number of observed resident handlings (the total number of physical caring activities such as lifting, turning and repositioning the resident) and (**B**) observed duration of care (the total time spend caring for the resident) on a whole day (day and evening shifts combined) according to the resident need-for-assistance scale.

**Figure 3 ijerph-19-10488-f003:**
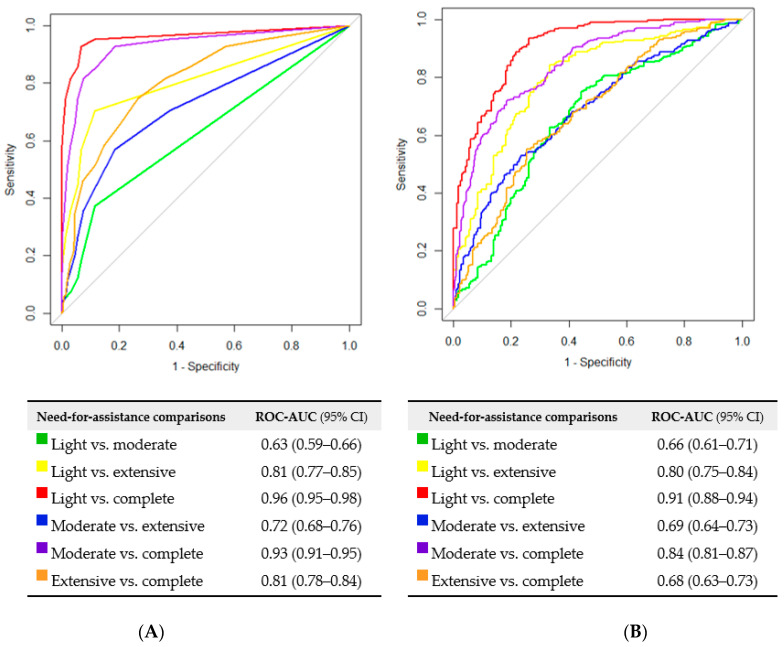
Receiver operating characteristic (ROC) curves of comparisons between the resident need-for-physical-assistance scale’s categories for (**A**) observed number of handlings and (**B**) observed duration of care. The tables show the comparison between the categories, the area under the ROC curve (ROC-AUC) value and 95% confidence intervals (95% CI) for each comparison.

**Table 1 ijerph-19-10488-t001:** Descriptive results of the residents’ observed need for physical assistance according to the resident need-for-physical-assistance scale.

	Resident Need-for-Assistance Classification				
	All Residents Mean ± SD	Light Physical Assistance Mean ± SD	Moderate Physical Assistance Mean ± SD	Extensive Physical Assistance Mean ± SD	Complete Physical Assistance Mean ± SD
	*n* = 1456	*n* = 290	*n* = 453	*n* = 262	*n* = 312
Resident body weight (kg)	65.46 ± 15.14	66.49 ± 15.49	64.85 ± 14.56	66.20 ± 15.86	64.78 ± 15.05
Whole day					
Total number of handling activities	3.2 ± 4.5	0.3 ± 0.9	0.9 ± 1.9	3.1 ± 3.7	8.1 ± 5.0
Number of lifting activities	1.0 ± 1.6	0.1 ± 0.3	0.3 ± 0.8	1.2 ± 1.6	2.3 ± 1.8
Number of repositioning activities	1.1 ± 1.8	0.2 ± 0.7	0.5 ± 1.0	1.2 ± 1.6	2.4 ± 3.4
Number of turning activities	1.1 ± 2.3	0.0 ± 0.2	0.2 ± 0.7	0.8 ± 1.8	3.3 ± 3.1
Number of handlings without support from the resident	2.3 ± 4.0	0.1 ± 0.6	0.4 ± 1.3	1.8 ± 3.2	6.5 ± 5.0
Duration of care (min)	46.9 ± 26.8	27.1 ± 18.5	36.9 ± 19.4	51.7 ± 24.2	67.8 ± 25.5

**Table 2 ijerph-19-10488-t002:** Spearman’s rho correlation coefficient between the resident need-for-assistance scale and observed outcomes, and area under the receiver operating characteristics curve (ROC-AUC) values.

	Spearman’s Rho (95% CI)	ROC-AUC *
Whole day		
Total number of handling activities	0.71 (0.67–0.74)	0.81
Number of lifting activities	0.63 (0.59–0.66)	0.74
Number of repositioning activities	0.52 (0.47–0.57)	0.71
Number of turning activities	0.61 (0.57–0.65)	0.71
Number of handlings without support from the resident	0.71 (0.67–0.73)	0.78
Duration of care (min)	0.57 (0.52–0.61)	0.76

* Not possible to obtain 95% CIs for the overall AUC value.

## Data Availability

The study datasets are available at the Danish National Archives, https://www.sa.dk/en/k/about-us (accessed on 15 July 2021).

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
