# Peer review of "A Simple Resident Need-for-Physical-Assistance Scale in Eldercare: Validation Using 4716 Observation Sequences of Caring Activities"

_ijerph, 2022, doi:10.3390/ijerph191710488_

Round 1

Reviewer 1 Report

Dear Authors,

I have read the manuscript and I send you my comments:

The purpose is of interest but the type of scale is note easy to understand. The authors must report what is the scale used, in which patients it can be used (elderly, with neurological diseases, with respiratory diseases, etc). The characteristic of the patients enrolled is missing as well as the age, the gender, etc  

Reviewer 4 Report

Thank you for giving me an opportunity to read the valuable paper. I believe that the paper is an interesting to the readers in geriatrics and gerontology. 

However, it is also necessary to make a major revision of the paper for the following causes:

Introduction

1. Overall, the authors insist that all the caregiver burden be equally distributed to individual caregivers. But I do not think that it is realistic. We have to consider the characteristics of caregivers such as age, sex, physical condition that could affect caregiver burden. 

2. The authors introduced their own rationale. More previous papers conducted in other countries should be introduced to attract international readers' attention. 

Material and Methods 

3. As for study population, it is said that the nursing homes were purposively selected. But, only one fourth of 83 nursing homes agreed to participate in the study. We need such an information on the recruitment. 

4. Did the authors adjust the results for the characteristics of the caregivers and/or the care receivers (older residents)? If not, it should be described in study limitations.

Discussion

5. I am still not sure how to apply the results to clinical settings. In Japan, similar results were used to decide care need level under the public long-term care insurance. I recommend to refer to previous studies to consider the clinical implications.   

Reviewer 5 Report

Please find comments and suggestion in attachment.

Round 2

Reviewer 4 Report

The paper is now suitable for the publication.

Author Response

Thank you for finding our manuscript suitable for publication. 

Reviewer 5 Report

Please find in attachment the review report

Round 3

Reviewer 5 Report

The manuscript has been sufficiently improved to warrant publication in IJERPH

Author Response

(The authors gave the same response as above.)
